

# Extension of the general unit hydrograph theory for the spread of salinity in estuaries

Huayang Cai[1,2] Bo Li[1,2], Junhao Gu[1,2], Tongtiegang Zhao[3], Erwan Garel[4]

[1]Institute of Estuarine and Coastal Research/State and Local Joint Engineering Laboratory of Estuarine Hydraulic Technology,
School of Ocean Engineering and Technology, Sun Yat-sen University, Guangzhou, 510275, China
[2]Guangdong Provincial Engineering Research Center of Coasts, Islands and Reefs/Southern Marine Science and Engineering
Guangdong Laboratory (Zhuhai), Zhuhai, 519082, China
[3]School of Civil Engineering, Sun Yat-sen University, Zhuhai, 519082, China
[4]Centre for Marine and Environmental Research (CIMA), University of Algarve, Faro, Portugal

*Correspondence to*: Erwan Garel (egarel@ualg.pt)

**Abstract.** From both practical and theoretical perspectives, it is essential to be able to express observed salinity distributions in terms of simplified theoretical models, which enable qualitative assessment to be made in many problems concerning water resources utilization (such as intake of fresh water) in estuaries. In this study, we propose a general and analytical salt intrusion model inspired by Guo's general unit hydrograph theory for predictions of flood hydrograph in a watershed. To derive a simple, general and analytical model of salinity distribution, we first make four hypotheses on the longitudinal salinity gradient based on empirical observations; we then derive a general unit hydrograph for the longitudinal salinity distribution in estuaries of the partial to well mixed type. The newly developed model can be well calibrated using a minimum of three salinity measurements along the estuary axis and does converge towards zero when distance approaches infinity asymptotically. The theory has been successfully applied to reproduce the salt intrusion in 21 estuaries worldwide, which suggests that the proposed method can be a useful tool for quickly assessing the spread of salinity under a wide range of riverine and tidal conditions and for quantifying the potential impacts due to human-induced and natural changes.

## 1 Introduction

An estuary is the place where the fresh water meets the saline water. It is crucial to quantify the spatial-temporal salinity dynamics determined by the competition between the advective salt flux due to river flow and the dispersive salt flux caused by tidal currents, since it directly affects water quality and the related water resources management in general. It is well known that the key to quantify the salinity distribution along an estuary is the efficient dispersion coefficient, which incorporates all mixing mechanisms that counteract the advective salt transport and regards the complex estuarine system as a whole. With the one-dimensional steady-state salt balance equation, indicating the equilibrium between the advective and dispersive transports of salt, it is possible to derive an empirical relationship for the salt intrusion in estuaries (Prandle, 1981; Savenije,1986, 1989, 1993, 2005, 2012; Lewis and Uncles, 2003; Gay and O' Donnell, 2007, 2009; Kuijper and Van Rijn, 2011; Cai et al., 2015; Zhang and Savenije, 2017, 2018). Amongst the proposed solutions, the empirical model using the Van der Burgh's coefficient





(e.g., Van der Burgh, 1972; Savenije, 1986) functions well in a wide range of estuaries worldwide (e.g., Savenije, 2005, 2012). In addition to practical applications, such an empirical model can be very useful from a physical perspective when its theoretical basis is well understood.

Recently, Guo (2022a, 2022b, 2022c) revisited the classical unit hydrograph (UH) theory, which is widely used in hydrology for predicting flood hydrograph from a known storm in a watershed. Based on three hypotheses on instantaneous UH derived from observations, he derived a general and analytical unit-volume hydrograph for S-hydrograph, which represents the discharge from a continuous excess rainfall occurring at a uniform rate for an indefinite period and is used to derive a UH of any storm duration. It appears that the shape of S-hydrographs is rather similar to the salinity distribution curve along estuaries,

while the instantaneous UH curve has similarity to the longitudinal salinity gradient, which opens the possibility that the UH method can be applied to describe the spread of salinity in estuaries.

The objective of this study is to derive a general and analytical expression of the salinity distribution and thus to derive the salinity gradient analytically following Guo's UH method (Guo, 2022a, 2022b, 2022c). To this end, we start with a review on Guo's general UH theory, together with the Savenije's empirical salt intrusion model, which is derived from the steady salt

balance equation and performs well against numerous salt measurements along many different estuaries (e.g., Savenije, 2005, 2012). Subsequently, we make four hypotheses based on empirical observations and follow the general UH theory, which leads to a newly developed analytical model for the spread of salinity in estuaries. The model was then applied to real estuaries with a wide range of riverine and tidal conditions. After that, the proposed model was compared with the conventional Savenije's model to discuss the physical foundation of the proposed model, which requires further study in the future.

**2 Review of the general unit hydrograph and empirical salt intrusion model**

**2.1 General unit hydrograph theory**

It was shown that a classical instantaneous UH $u(t)$ [T-1] (representing the discharge due to a unit-volume input of excess rainfall) with regard to time $t$ [T] should satisfy the following properties (Chow et al., 1988):

$$u(t) = 0 \quad \text{for } t \leq 0, \tag{1}$$

$$0 \leq u(t) \leq u_p \quad for\ t > 0, \tag{1}$$

$$u(t) \to 0 \quad for\ t \to \infty, \tag{2}$$

and

$$\int_0^\infty u(t)\, d\, t = 1 \tag{3}$$

where $u_p$ [T-1] in Eq. (2) represents the peak discharge of instantaneous UH. Eq. (4) is the mass conservation equation,

indicating that the total outflow volume (i.e., the left side) corresponds to the unit-volume input (i.e., the right side). Making





use of the definition of the instantaneous UH $u(t) = \mathrm{d}U/\mathrm{d}t$ (where $U$ represents the dimensionless S-hydrograph), Eq. (4) can be rewritten as:

$$U(\infty) = 1 \tag{4}$$

Nash (1957) derived an analytical expression for the instantaneous UH that satisfies Eqs. (1)-(4), which is the well-known

Nash's gamma function:

$$u(t) = \frac{1}{\alpha_1 \Gamma(\alpha_2)} \left( \frac{t}{\alpha_1} \right)^{\alpha_2 - 1} exp(-t/\alpha_1) \tag{5}$$

where $\alpha 1$ and $\alpha 2$ are model parameters, while $\Gamma(\alpha 2)$ is the gamma function. Figure 1 illustrates an arbitrary distribution ($t$=0-30) of the instantaneous UH for given values of $\alpha 1 = \alpha 2 = 3$. It can be seen from Figure 1 that the instantaneous UH consists of two distinct regions: the rising limb described by the power function in Eq. (6) and the recessing limb described by the

70 exponential function in Eq. (6).

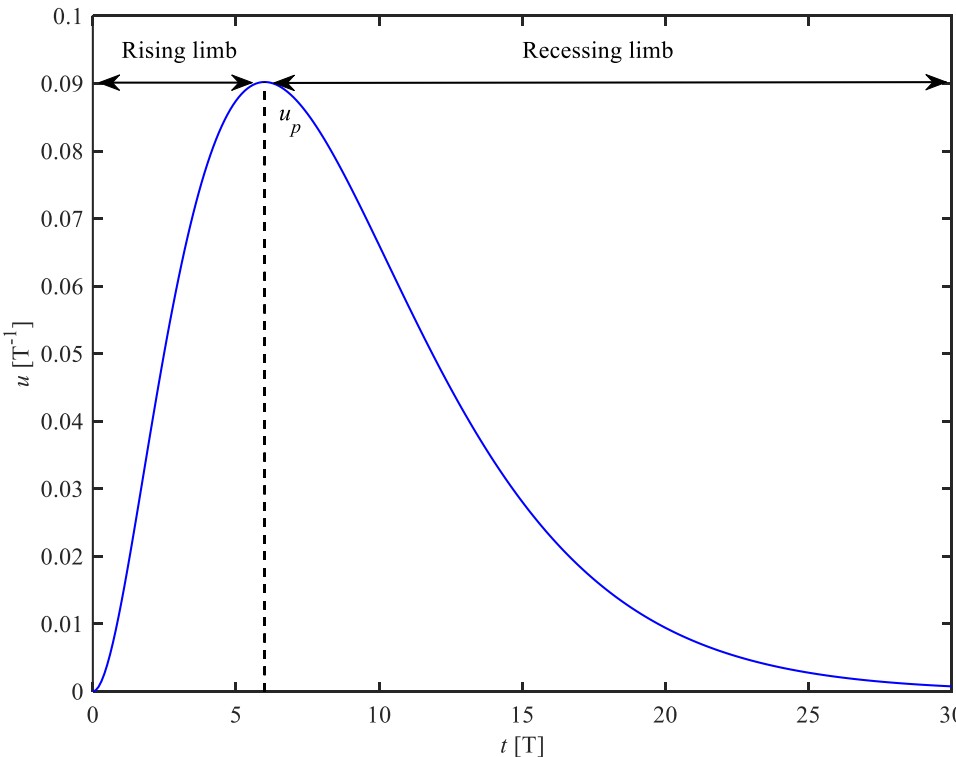

**Figure 1. Illustration of the instantaneous UH analytically computed using Eq. (5) for given values of $\alpha_1 = \alpha_2 = 3$.**

Although Eq. (6) is widely used as the analytical solution of instantaneous UH, it has two weaknesses: (1) it has a zero initial condition, which is not necessary the case for real instantaneous UH; and (2) a general and analytical solution of S-hydrograph

does not exist. In order to remove these weaknesses, Guo (2022a) made three hypotheses on the instantaneous UH based on empirical observations: (1) the instantaneous UH increases exponentially along the rising limb; (2) the instantaneous UH





decreases exponentially along the recessing limb; (3) the instantaneous UH tends to 0 and the S-hydrograph tends to 1 as t tends to infinity. Subsequently, he derived a general and analytical solution for S-hydrograph:

$$U(t) = 1 - \left\{1 + \beta_2 \, exp[\beta_1(t/t_p) - 1]\right\}^{-1/\beta_2} \tag{6}$$

where $\beta_1$ is the dimensionless rising coefficient determined by the watershed characteristics, $\beta_2$ is the dimensionless recessing coefficient affected by the downstream water surface condition, while $t=t_p$ corresponds to the inflection point with maximum instantaneous UH:

$$\left.\frac{d\,u}{d\,t}\right|_{t=t_p} = \left.\frac{d^2\,U}{d\,t^2}\right|_{t=t_p} = 0 \tag{7}$$

With Eq. (7), the instantaneous UH $u(t)$ is expressed as:

$$u(t) = \frac{\beta_1 \, exp[\beta_1(t/t_p)-1]\{1+\beta_2 \, exp[\beta_1(t/t_p)-1]\}^{-(1+1/\beta_2)}}{t_p} \tag{8}$$

To illustrate the general and analytical solutions of S-hydrograph from Eq. (6) and instantaneous UH from Eq. (8), Figure 2 shows the computed $U(t)$ (solid line) and $u(t)$ (dashed line) for given values of $\beta_1=\beta_2=3$ and $t_p=10$.

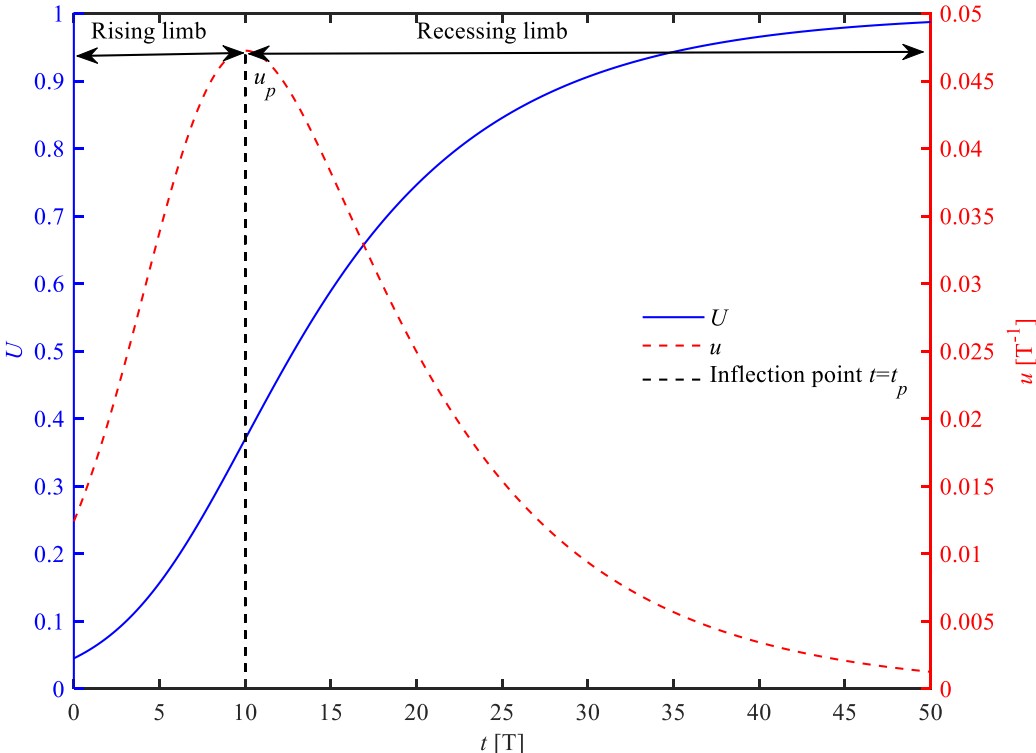

**Figure 2. Illustration of the S-hydrograph and instantaneous UH analytically computed using Eqs. (6) and (8), respectively, for given values of $\beta_1=\beta_2=3$ and $t_p=10$.**





## 2.2 Savenije's salt intrusion model

In estuaries, the key to derive an empirical relationship for the salinity distribution is the dispersion coefficient, which is either constant (e.g., Gay and O' Donnell, 2007) or variable (e.g., Van der Burgh, 1972; Prandle, 1981). Based on the effective tidal average dispersion under steady state conditions, Van der Burgh (1972) proposed an empirical relationship for the dispersion

coefficient:

$$\frac{\partial D}{\partial x} = -K \frac{|Q|}{A} \tag{9}$$

where $D$ [$L^2T^{-1}$] is the longitudinal dispersion coefficient, $x$ [L] is the longitudinal coordinate measured in landward direction, $Q$ [$L^3T^{-1}$] is the fresh water discharge, $A$ [$L^2$] the tidally averaged cross-sectional area, and $K$ is the dimensionless Van der Burgh coefficient.

It is assumed that the longitudinal cross-sectional area follows an exponential function:

$$A = A_0 \, exp(-x/a) \tag{10}$$

where $A_0$ is the cross-sectional area at the estuary mouth and $a$ is the convergence length. Integration of Eq. (9) and taking into account the exponential variation of the cross-sectional area using Eq. (10) yields an analytical description of the longitudinal effective dispersion (Savenije, 2005, 2012):

$$\frac{D}{D_0} = 1 - \frac{Ka|Q|}{D_0 A_0}[exp(x/a) - 1] \tag{11}$$

where $D_0$ is the dispersion coefficient at the estuary mouth.

With Eq. (10), Savenije (2005, 2012) derived a one-dimensional empirical model for salt intrusion based on the tidally averaged cross-sectional mass conservation equation:

$$F = -|Q|S - AD\frac{\partial S}{\partial x} \tag{12}$$

where $F$ [$MT^{-1}$] and $S$ [$ML^{-3}$] are the tidally averaged salt flux and salinity, respectively.

In a steady state situation with no net salt flux (i.e., $F=0$), Eq. (12) can be rewritten as:

$$\frac{d\,S}{S} = -\frac{|Q|}{AD} d\,x \tag{13}$$

We can combine Eqs. (10) and (14) into a general relationship between the dispersion coefficient and salinity through the Van der Burgh's coefficient (Savenije, 2005, 2012):

$$D/D_0 = (S/S_0)^K \tag{14}$$

where $S_0$ is the salinity concentration at the estuary mouth.

Combing Eqs. (12) and (15) yields the tidally averaged salinity along an estuary (Savenije , 2005, 2012):

$$\frac{S}{S_0} = \left(1 - \frac{Ka|Q|}{D_0 A_0}[exp(x/a) - 1]\right)^{1/K} \tag{15}$$

To make Eq. (15) dimensionless, we introduce the following dimensionless parameters:





$$S^* = \frac{S}{S_0}, \quad \gamma = \frac{c_0}{\omega a}, \quad D^* = \frac{|Q|c_0}{D_0 A_0 \omega}, \quad x^* = \frac{x\omega}{c_0}, \tag{16}$$

where $S^*$ is dimensionless salinity that is scaled by the value at the estuary mouth, $\gamma$ represents the estuary shape number describing the convergence rate of an estuary, $\omega$ is the tidal frequency, $D^*$ is the dimensionless dispersion coefficient representing the downstream dispersion condition, $x^*$ is the dimensionless longitudinal coordinate that is normalized by the frictionless wavelength in prismatic channels and $c_0$ is the classical wave celerity of a frictionless progressive wave, which is

125 defined as:

$$c_0 = \sqrt{gh/r_S} \tag{17}$$

in which $g$ [LT$^{-2}$] is the acceleration due to gravity, $h$ [L] the tidally averaged depth, and $r_S$ the storage width ratio (see Savenije et al., 2008). Here the asterisk denotes a dimensionless variable.

Thus, Eq. (16) can be rearranged as (Cai et al., 2015):

$$S^* = \left(1 - \frac{D^* K}{\gamma}\left[exp\left(x^* \gamma\right) - 1\right]\right)^{1/K} \tag{18}$$

With Eq. (18), it is possible to derive an analytical expression for the salt intrusion length (i.e., the distance from the estuary mouth to the location where the water is totally fresh), which is obtained by setting $S^*=0$ in Eq. (18):

$$L^* = \frac{1}{\gamma} ln\left(\frac{\gamma}{D^* K} + 1\right) \tag{19}$$

## 3 General unit hydrograph theory for salt intrusion

Suppose there is an ocean coupling to an estuary with coordinate origin located at the estuary mouth. If a unit-volume of excess salinity from the ocean is locally ($\Delta x \rightarrow 0$) released into the estuary during a time required for an equilibrium to occur, the resulting hydrograph is the instantaneous UH $dS^*(x)/dx$ that corresponds to the S-hydrograph $S^*(x)$ in dimensionless form. Similar to Guo's UH method (Guo, 2022a), we make four hypotheses on the instantaneous UH (representing the instantaneous rate of change with respect to the salinity at a specific position along the estuary axis, i.e., salinity gradient) for salinity

distribution based on depth-average observations along estuaries (i.e., data facts):

Hypothesis 1: Along the recessing limb, the salinity gradient $dS^*(x)/dx$ decreases exponentially, which makes the salinity $S^*(x)$ to decay exponentially in a convex shape.

Hypothesis 2: The salinity is scaled by the almost constant salinity in the deep ocean, i.e., approximately 36 kg/m$^3$; Thus, as $x$ tends to negative infinity, the salinity gradient tends to zero and the salinity $S^*(x)$ tends to 1.

Hypothesis 3: Along the rising limb, the salinity gradient $dS^*(x)/dx$ increases exponentially, which makes the salinity $S^*(x)$ to decay exponentially in a concave shape.

Hypothesis 4: As $x$ tends to infinity, the salinity gradient tends to zero and the salinity $S^*(x)$ tends to 0.



It should be noted that the above hypotheses 1 and 3 are in principle valid only for well mixed or partially mixed estuaries, where salt intrusion really matters. From a practical perspective, this is not a restrictive assumption since the salt wedge in highly stratified condition only occurs at the time of high river discharge, when flood protection is generally the main concern and salt intrusion is not relevant (Savenije, 2005, 2012).

According to the first hypothesis, along the recessing limb, $dS^*(x)/dx$ and $S^*(x)$ satisfy the following relationship:

$$\frac{d S^*}{d x} = -\mu S^* \tag{20}$$

where $\mu$ represents the recessing coefficient [L$^{-1}$]. Meanwhile, the second hypothesis requires that $dS^*(x)/dx=0$ and $S^*(x)=1$ (representing the constant salinity in the ocean) at $x\rightarrow-\infty$. To meet this requirement, we revise Eq. (20) as:

$$\frac{d S^*}{d x} = -\mu\left(1 - S^*\right) \tag{21}$$

because $S^*(-\infty)=1$. Integrating Eq. (21) for $S^*(x)$ and applying the initial condition (i.e., $x=0$, $S^*=S_0^*$) results in:

$$S^* = 1 + \left(S_0^* - 1\right) exp(\mu x) \tag{22}$$

Similarly, according to the third hypothesis, along the rising limb, we have:

$$\frac{d S^*}{d x} = -\frac{\mu}{m} S^* \tag{23}$$

where $m$ represents the dimensionless rising coefficient. Integrating Eq. (23) for $S^*(x)$ and applying the initial condition (i.e., $x=0$, $S^*=S_0^*$) results in:

$$S^* = S_0^* exp\left(-\frac{\mu}{m} x\right) \tag{24}$$

which satisfies the fourth hypothesis. We can combine Eqs. (22) and (24) into a generalized differential equation:

$$\frac{d S^*}{d x} = -\frac{\mu}{m} S^*\left(1 - S^{*m}\right) \tag{25}$$

which reduces to Eq. (21) at $x\rightarrow-\infty$ where $S^*=1$ and Eq. (23) at $x\rightarrow\infty$ where $S^*\rightarrow0$. The inflection point $x=x_p$ where

$$\left.\frac{d^2 S^*}{d x^2}\right|_{x=x_p} = 0 \tag{26}$$

corresponds to the maximum absolute value of $dS^*/dx$ (i.e., maximum salinity gradient). Integrating Eq. (25) for $S^*(x)$ and applying Eq. (26) result in:

$$S^* = \left\{1 + m\, exp\left[\mu\left(x - x_p\right)\right]\right\}^{-1/m} \tag{27}$$

To make $\mu$ dimensionless, we make a transform $\mu x_p\rightarrow\mu$, then Eq. (27) can be revised as:

$$S^* = \left\{1 + m\, exp\left[\mu\left(x/x_p - 1\right)\right]\right\}^{-1/m} = \left\{1 + m\, exp\left[\mu\left(x^* - 1\right)\right]\right\}^{-1/m} \tag{28}$$

where $x^*=x/x_p$ is the dimensionless distance scaled by the position of inflection point $x_p$. With Eq. (28), the instantaneous UH $dS^*/dx$ is written as:





$$\frac{dS^*}{dx^*} = -\mu \, exp[\mu(x^*-1)]\{1 + m \, exp[\mu(x^*-1)]\}^{-(1+1/m)}$$
(29)

which satisfies the general UH definition, i.e., $\int_{-\infty}^{+\infty} \frac{dS^*}{dx^*} = 1$.

It can be seen from Eq. (28) that the theoretical salt intrusion length $L^*$ is not available since $S^*$ tends to 0 as $x$ approaches infinity asymptotically. However, it is possible to define a specific salt intrusion length for a given salinity threshold $S_f^*$ (such as 0.01) by substituting $S_f^*$ into Eq. (28):

$$L^* = \frac{1}{\mu} ln\left(\frac{S_f^{*-m}-1}{m}\right) + 1$$
(30)

Figure 3 illustrates the spatial distribution of the S-hydrograph (salinity distribution $S^*$) and its instantaneous UH (salinity gradient $dS^*/dx^*$) for given values of $\mu=1.5$ and $m=1$.

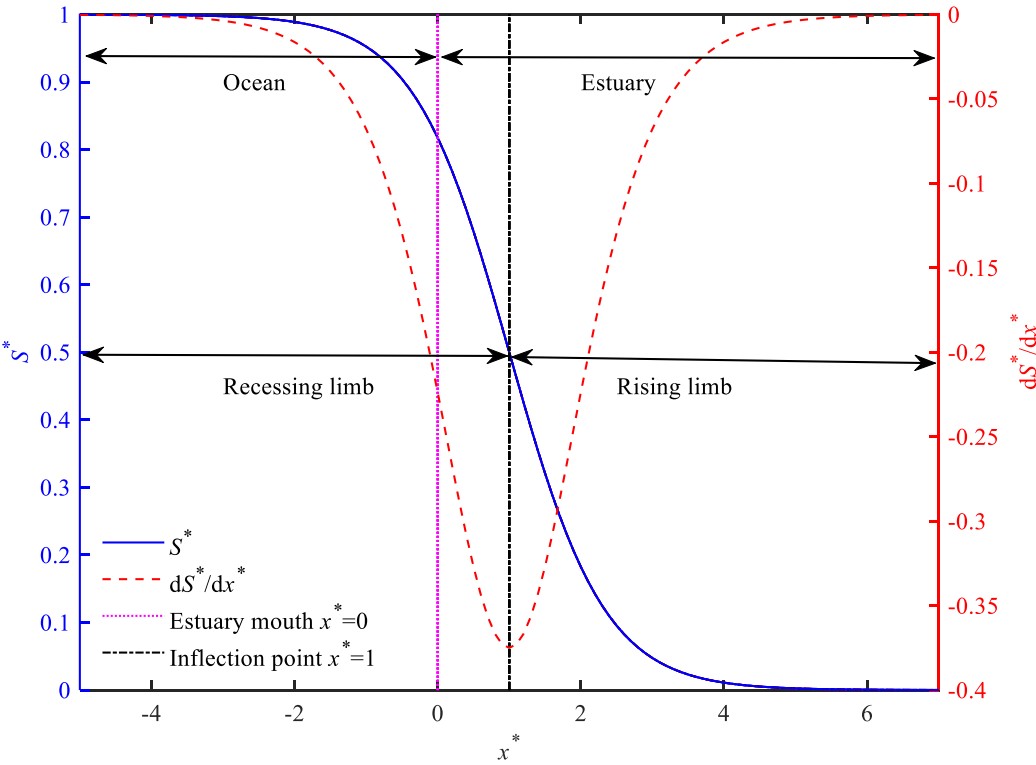

**Figure 3. S-hydrograph $S^*$ (salinity concentration) and its instantaneous UH $dS^*/dx^*$ (salinity gradient) as a function of dimensionless**
**distance $x^*$ for given values of $\mu=1.5$, $m=1$.**





## 4 Results and discussion

### 4.1 Sensitivity analysis of the proposed salt intrusion model

Although Eqs. (28), (29) and (30) are analytical, the sensitivity to the two controlled parameters ($\mu$ and $m$) is not straightforward and directly clear. Thus, it is worthwhile to have a sensitivity analysis on the two calibrated parameters. Figure
4 presents the analytical solutions of the longitudinal salinity and its gradient as a function of $\mu$ and $m$. It can be clearly seen from Figure 4 that two distinct regions of estuaries display very different behaviour. For $x^*<1$ we see an exponential increase of salinity gradient until a minimum value is reached at a critical position $x=x_p$ (or $x^*=1$) defined by Eq. (26), beyond which the salinity gradient decreases exponentially until zero is reached asymptotically. The sensitivity analysis shows that the recessing coefficient $\mu$ determines the change rate of both the rising and recessing limbs (Figures 4a, c). With regard to the
rising coefficient $m$, it can be seen from Figure 4b that the coefficient $m$ exerts more influence along the rising limb and thus affects mainly the salinity distribution after the inflection point (Figures 4b, d). In addition, Figures 4c, d show that $m=1$ gives a symmetric salinity gradient about $x=x_p$ ($x^*=1$); $0 \leq m < 1$ gives a negatively skewed salinity gradient; and $m>1$ gives a positively skewed salinity gradient.

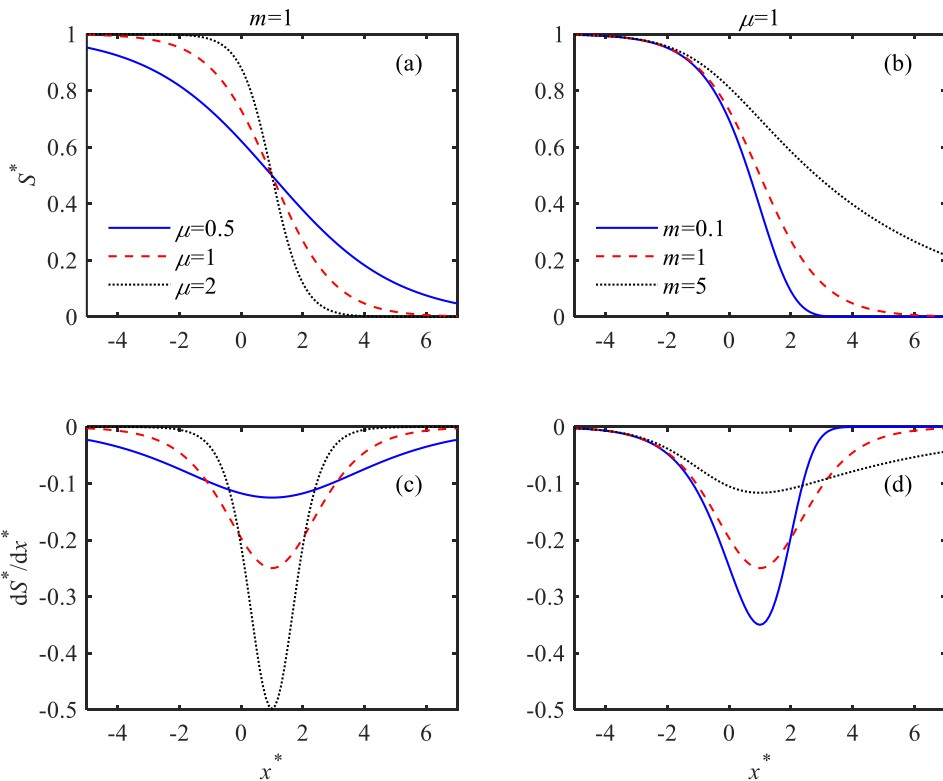

**Figure 4. Sensitivity analysis of the dimensionless salinity $S^*$ and its gradient d$S^*$/d$x$ with regard to the recessing coefficient $\mu$ and the rising coefficient $m$.**



To understand the response of the salt intrusion length to both calibrated parameters, Eq. (30) was used to analytically compute $L^*$ for a wide range of $\mu$ and $m$ values (Figure 5) considering a salinity threshold $S_f^*=0.01$. We can clearly see that the isolines are almost linear, converging towards the origin of the $m$-$\mu$ diagram. Generally, the salt intrusion length increases with $m$,

while it decreases with $\mu$. This suggests that the model parameter $m$ is generally proportional to the strength of tidal dynamics that induces dispersive transport of salt in the landward direction, while the model parameter $\mu$ is proportional to the strength of the riverine flushing seaward. With this plot, it is possible to understand the potential impacts of different hydrodynamic conditions on the salt intrusion length, which is particularly useful for decision makers for salt intrusion prevention.

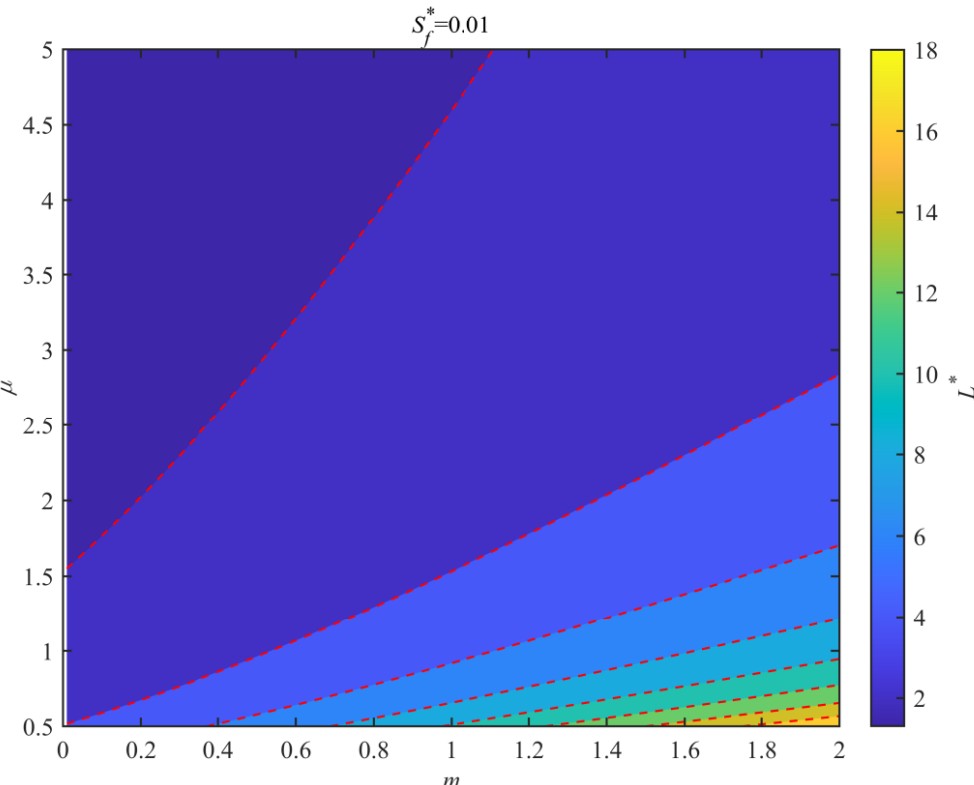

**Figure 5. Response of salt intrusion length $L^*$ to the dimensionless parameters $\mu$ and $m$ for a given salinity threshold $S_f^*=0.01$.**

## 4.2 Application to real estuaries

The proposed salt intrusion model has been applied to observations in real estuaries worldwide with a wide range of different riverine and tidal hydrodynamics. In Table 1, a selection is presented of 21 estuaries, of which 89 salt intrusion measurements collected at either high water slack (HWS) or low water slack (LWS) are available (all observations are available on the web

at https://salinityandtides.com/data-sources/). In this table the three model parameters $x_p$, $\mu$ and $m$ were obtained by fitting Eq. (28) to the observed longitudinal salt intrusion. This can be easily done by means of nonlinear curve-fitting method in least





squares sense (such as using the Matlab 'lsqcurvefit.m' function). The model performance was evaluated by using the root mean square error (RMSE). Figure 6 shows the comparison of observed and computed salinity concentration in different estuaries. It can be seen from Figure 6 that the correspondence with the observed salt intrusion is good with the RMSE being

1.1 kg/m$^3$ on average (see also the model performance reported in Table 1 for many estuaries worldwide with distinct salt intrusion lengths).

**Table 1. Measured salinity distributions, calibrated parameters, computed salt intrusion length and the model performance in terms of RMSE**

| Estuary | Date | Condition | $x_p$ (km) | $\mu$ | $m$ | $L$ (km) | RMSE (kg/m$^3$) |
|---|---|---|---|---|---|---|---|
| Bernam | 01/06/12 | HWS | 22.73 | 2.17 | 0.47 | 51.97 | 0.18 |
| Bernam | 01/06/12 | LWS | 10.41 | 1.07 | 0.36 | 34.44 | 0.39 |
| Chaophy | 05/06/62 | HWS | 23.61 | 1.47 | 0.32 | 61.44 | 0.71 |
| Chaophy | 05/06/62 | LWS | 2.49 | 0.13 | 0.10 | 35.48 | 0.40 |
| Chaophy | 23/02/82 | HWS | 19.99 | 1.23 | 0.10 | 48.69 | 0.56 |
| Chaophy | 29/01/83 | HWS | 28.08 | 1.72 | 0.17 | 59.86 | 0.42 |
| Chaophy | 16/01/87 | HWS | 5.68 | 0.38 | 0.10 | 32.05 | 1.04 |
| Corantijn | 09/12/78 | HWS | 20.43 | 1.04 | 0.45 | 73.81 | 0.21 |
| Corantijn | 09/12/78 | LWS | 11.05 | 3.30 | 3.38 | 59.09 | 0.10 |
| Corantijn | 14/12/78 | HWS | 13.80 | 0.66 | 0.60 | 81.26 | 0.01 |
| Corantijn | 14/12/78 | LWS | 6.63 | 1.31 | 2.69 | 64.51 | 0.22 |
| Corantijn | 20/12/78 | HWS | 15.27 | 0.52 | 0.10 | 66.90 | 0.21 |
| Corantijn | 20/12/78 | LWS | 2.99 | 0.61 | 2.87 | 62.63 | 0.18 |
| Elbe | 09/07/02 | HWS | 0.01 | 0.00 | 0.10 | 44.03 | 0.58 |
| Elbe | 04/04/04 | HWS | 10.21 | 0.31 | 0.10 | 67.82 | 0.90 |
| Elbe | 21/09/04 | HWS | 35.77 | 1.18 | 0.10 | 89.37 | 0.88 |
| Elbe | 21/09/04 | LWS | 14.36 | 0.46 | 0.15 | 73.98 | 0.65 |
| Endau | 28/03/13 | HWS | 14.81 | 2.16 | 0.10 | 26.89 | 1.17 |
| Endau | 28/03/13 | LWS | 3.83 | 0.46 | 0.10 | 18.64 | 0.82 |
| Incomati | 05/09/82 | HWS | 20.61 | 4.41 | 1.80 | 56.48 | 0.32 |
| Incomati | 23/06/93 | HWS | 16.62 | 3.94 | 1.43 | 42.78 | 0.66 |
| Incomati | 23/06/93 | LWS | 9.46 | 1.52 | 0.78 | 33.21 | 0.27 |
| Incomati | 07/07/93 | HWS | 16.56 | 3.88 | 1.56 | 45.37 | 0.27 |
| Incomati | 07/07/93 | LWS | 8.56 | 2.01 | 1.41 | 34.60 | 0.72 |
| Kurau | 27/02/13 | HWS | 8.78 | 1.89 | 0.10 | 16.98 | 1.72 |
| Kurau | 27/02/13 | LWS | 0.00 | 0.00 | 0.10 | 5.54 | 0.86 |



| | | | | | | | |
|---|---|---|---|---|---|---|---|
| Kurau | 28/02/13 | HWS | 9.06 | 1.97 | 0.10 | 17.17 | 1.40 |
| Kurau | 28/02/13 | LWS | 0.00 | 0.00 | 0.10 | 6.29 | 0.42 |
| Lalang | 20/10/89 | HWS | 24.60 | 1.28 | 0.10 | 58.50 | 2.64 |
| Lalang | 20/10/89 | LWS | 0.00 | 0.00 | 0.10 | 17.02 | 1.96 |
| Landak | 15/09/09 | HWS | 15.93 | 0.57 | 0.10 | 65.20 | 0.71 |
| Landak | 15/09/09 | LWS | 7.18 | 0.35 | 0.10 | 42.94 | 0.39 |
| Limpopo | 04/04/80 | LWS | 3.17 | 0.40 | 0.10 | 17.31 | 0.70 |
| Limpopo | 31/12/82 | HWS | 39.06 | 2.13 | 0.10 | 71.49 | 0.36 |
| Limpopo | 31/12/82 | LWS | 30.23 | 1.49 | 0.10 | 66.16 | 0.27 |
| Limpopo | 14/07/94 | HWS | 24.73 | 1.59 | 0.10 | 52.18 | 0.67 |
| Limpopo | 24/07/94 | HWS | 20.67 | 3.29 | 2.01 | 74.36 | 1.45 |
| Limpopo | 24/07/94 | LWS | 12.22 | 2.26 | 2.76 | 75.31 | 1.92 |
| Limpopo | 10/08/94 | HWS | 16.65 | 5.26 | 6.15 | 100.58 | 1.24 |
| Limpopo | 10/08/94 | LWS | 13.78 | 1.96 | 2.28 | 82.03 | 1.26 |
| Maeklong | 09/04/77 | HWS | 15.65 | 2.01 | 0.21 | 31.48 | 1.16 |
| Maeklong | 09/04/77 | LWS | 7.65 | 0.91 | 0.10 | 22.52 | 1.34 |
| Maputo | 28/04/82 | HWS | 18.65 | 4.09 | 0.30 | 29.08 | 0.69 |
| Maputo | 28/04/82 | LWS | 2.20 | 0.90 | 2.04 | 23.52 | 0.00 |
| Maputo | 15/07/82 | HWS | 18.07 | 5.05 | 1.77 | 45.25 | 0.22 |
| Maputo | 19/04/84 | HWS | 15.09 | 2.18 | 0.10 | 27.32 | 0.64 |
| Maputo | 19/04/84 | LWS | 4.25 | 1.93 | 0.90 | 13.63 | 0.05 |
| Maputo | 17/05/84 | HWS | 16.78 | 3.18 | 0.16 | 26.91 | 0.90 |
| Maputo | 17/05/84 | LWS | 1.55 | 2.45 | 5.25 | 15.77 | 2.13 |
| Maputo | 29/05/84 | HWS | 6.14 | 1.09 | 0.25 | 18.31 | 0.32 |
| Maputo | 29/05/84 | LWS | 18.59 | 2.66 | 0.10 | 30.92 | 1.20 |
| Muar | 01/08/12 | HWS | 11.09 | 1.22 | 0.66 | 41.90 | 0.50 |
| Muar | 01/08/12 | LWS | 0.50 | 0.05 | 0.53 | 30.05 | 0.24 |
| Pangani | 27/10/07 | HWS | 21.27 | 3.61 | 0.10 | 31.66 | 2.14 |
| Pangani | 27/10/07 | LWS | 8.78 | 1.32 | 0.10 | 20.50 | 0.63 |
| Pangani | 11/12/07 | HWS | 16.89 | 3.37 | 0.10 | 25.74 | 1.67 |
| Pangani | 11/12/07 | LWS | 6.47 | 1.11 | 0.10 | 16.72 | 0.51 |
| Perak | 13/03/13 | HWS | 5.02 | 1.01 | 0.81 | 24.61 | 0.40 |
| Pungue | 26/09/80 | HWS | 55.28 | 2.93 | 0.10 | 88.64 | 0.63 |



| | | | | | | | |
|---|---|---|---|---|---|---|---|
| Pungue | 26/05/82 | HWS | 33.10 | 2.10 | 0.10 | 60.99 | 0.39 |
| Pungue | 06/08/82 | HWS | 39.44 | 2.87 | 0.10 | 63.72 | 0.36 |
| Pungue | 06/08/82 | LWS | 23.37 | 0.96 | 0.10 | 66.15 | 0.67 |
| Pungue | 22/09/82 | HWS | 46.59 | 3.34 | 0.10 | 71.24 | 0.54 |
| Pungue | 22/09/82 | LWS | 29.20 | 1.41 | 0.10 | 65.74 | 0.75 |
| Pungue | 29/10/82 | LWS | 23.77 | 1.73 | 0.10 | 47.98 | 0.01 |
| Pungue | 03/10/93 | HWS | 61.29 | 4.71 | 0.10 | 84.26 | 1.70 |
| Pungue | 12/10/93 | HWS | 54.61 | 6.09 | 0.10 | 70.44 | 0.63 |
| Pungue | 12/10/93 | LWS | 39.86 | 3.21 | 0.10 | 61.82 | 1.59 |
| Pungue | 16/10/93 | HWS | 74.46 | 7.52 | 0.10 | 91.96 | 0.68 |
| Pungue | 16/10/93 | LWS | 54.74 | 3.16 | 0.10 | 85.32 | 0.82 |
| Pungue | 31/01/02 | HWS | 17.77 | 1.16 | 0.25 | 50.52 | 0.58 |
| Pungue | 27/02/02 | HWS | 13.26 | 1.54 | 1.14 | 57.38 | 1.00 |
| Pungue | 27/02/02 | LWS | 1.88 | 0.26 | 0.61 | 25.62 | 0.38 |
| Pungue | 01/03/02 | HWS | 17.54 | 1.26 | 0.76 | 69.29 | 0.62 |
| Selangor | 01/08/12 | HWS | 10.48 | 2.11 | 0.10 | 19.25 | 1.52 |
| Selangor | 01/08/12 | LWS | 0.00 | 0.00 | 0.10 | 6.13 | 3.18 |
| Sinnamary | 12/11/93 | HWS | 5.67 | 1.95 | 0.10 | 10.80 | 0.89 |
| Sinnamary | 27/04/94 | HWS | 5.75 | 1.32 | 0.10 | 13.42 | 1.34 |
| Sinnamary | 02/11/94 | HWS | 9.06 | 2.53 | 0.10 | 15.38 | 1.56 |
| Sinnamary | 02/11/94 | LWS | 0.88 | 0.23 | 0.10 | 7.51 | 0.29 |
| Sinnamary | 03/11/94 | HWS | 7.49 | 1.77 | 0.10 | 14.99 | 1.23 |
| Tha-chin | 16/04/81 | HWS | 22.45 | 1.00 | 0.10 | 62.00 | 0.62 |
| Tha-chin | 27/02/86 | HWS | 16.88 | 1.77 | 0.63 | 48.47 | 0.01 |
| Tha-chin | 01/03/86 | HWS | 18.79 | 2.19 | 0.85 | 53.64 | 0.02 |
| Tha-chin | 13/08/87 | HWS | 15.90 | 1.11 | 0.10 | 41.27 | 0.72 |
| Tha-chin | 13/08/87 | LWS | 4.77 | 2.22 | 2.25 | 25.32 | 0.16 |
| Thames | 07/04/49 | LWS | 43.41 | 1.82 | 0.10 | 85.55 | 0.73 |
| Westerschelde | 02/11/00 | HWS | 82.23 | 2.93 | 0.10 | 131.81 | 0.84 |
| Westerschelde | 02/11/00 | LWS | 73.90 | 2.41 | 0.10 | 128.11 | 0.99 |



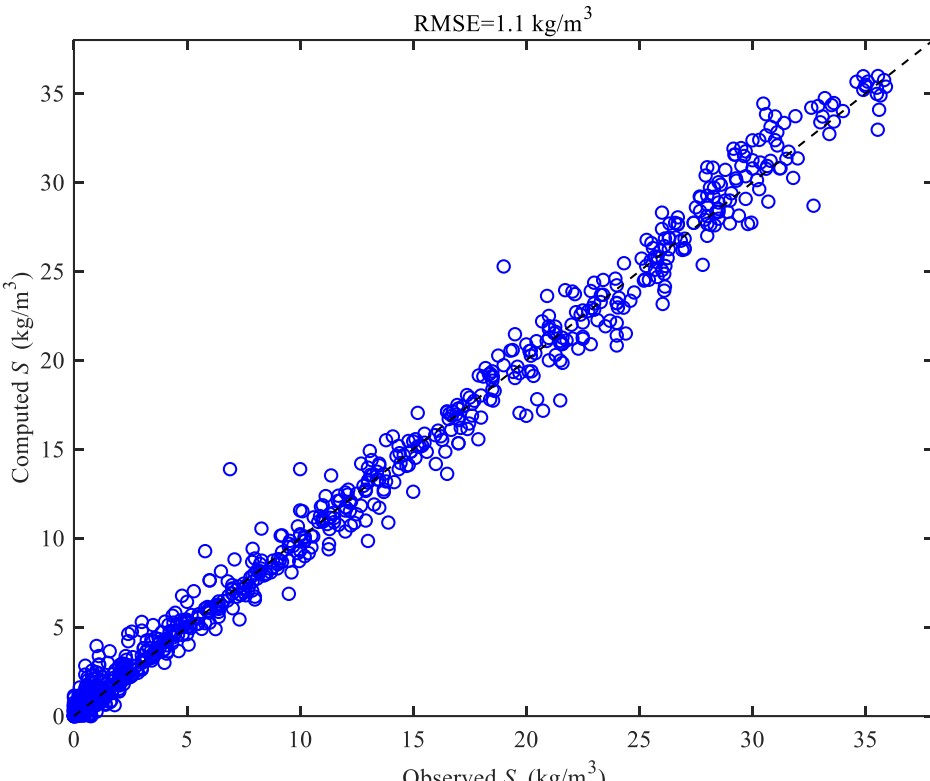

**Figure 6. Comparison between analytically computed salinity concentrations and 89 observations in 21 estuaries worldwide.**

For illustrations, Figure 7 shows the longitudinal computation applied to the Pungue, Incomati and Limpopo estuaries by means of both the newly proposed and Savenije's models. Generally, the results of the two models are satisfactory for the different shapes of salt intrusion curves in well mixed or partially mixed estuaries (Savenije, 2005, 2012): (1) a "dome-shape"

intrusion curve (such as Pungue, Figure 7a), which generally occurs in strong funnel-shaped estuaries; (2) a "bell-shaped" intrusion curve (such as Incomati, Figure 7b), which generally occurs in estuaries that have a trumpet shape ; (3) a "recession-shape" intrusion curve (such as Limpopo, Figure 7c), which generally occurs in narrow estuaries with a near-prismatic shape and a high river discharge. It can be seen from Figure 7 that one important difference of performance between these two models lies in the rising limb when the distance approaches infinity. As $x$ tends to infinity, the salinity gradient of the newly proposed

model asymptotically approaches to zero, while it reduces to zero at a critical position corresponding to the salt intrusion length from Eq. (19) for Savenije's model. This feature allows an improved fit with observations at the toe of the salt intrusion curve (e.g., Figure 7a). Figures S1-S8 show the comparison between the observed longitudinal salinity and the analytically computed salt intrusion curves along 21 estuaries worldwide (see the Supplementary Material).



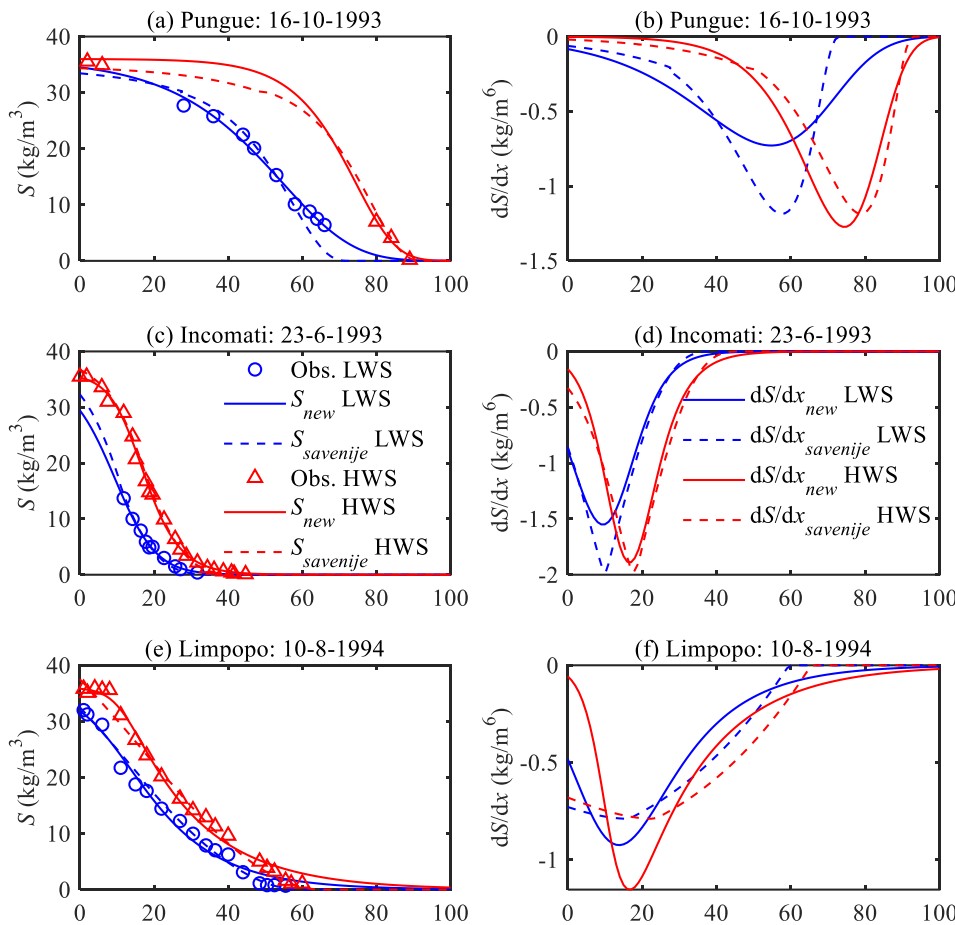

**Figure 7. Observed and analytically computed salt intrusion curves using the newly proposed and Savenije's models in the Pungue estuary (a, b), in the Incomati estuary (c, d) and in the Limpopo estuary (e, f).**

**4.3 Analytical difference with Savenije's salt intrusion model**

In order to understand the differences between the newly proposed model and empirical solutions based on the steady salt balance equation, we have made a comparison with the widely used Savenije's salt intrusion model (Savenije, 2005, 2012) by means of a Taylor expansion. The Taylor expansion of Savenije's salt intrusion model, i.e., Eq. (18), is written as:

$$S^* = 1 - D^* x^* + \frac{1}{2}\left(D^{*2} - D^* \gamma - D^{*2} K\right) x^{*2} + O\left(x^{*3}\right) \tag{31}$$

To make a comparison of the proposed model (i.e., Eq. (28)) with Eq. (18), we introduce $-m \rightarrow K$ (the Van der Burgh's coefficient), then Eq. (28) becomes:

$$S^* = \left\{1 - K\,exp\left[\mu\left(x^* - 1\right)\right]\right\}^{1/K} \tag{32}$$



The Taylor expansion of Eq. (32) is written as:

$$S^* = (1 - Ke^{-\mu})^{1/K} - (1 - Ke^{-\mu})^{(1/K-1)}e^{-\mu}\mu x^* + \frac{1}{2}(1 - Ke^{-\mu})^{(1/K-2)}e^{-\mu}\mu^2(e^{-\mu} - 1)x^{*2} + O(x^{*3}) \tag{33}$$

It is difficult to compare directly Eq. (31) and Eq. (33). Alternatively, the difference between Eq. (18) and Eq. (32) can be explored by looking at the exponential function parts, which can be expanded by the Taylor series:

$$\frac{D^*}{\gamma}[exp(x^*\gamma) - 1] = D^*x + \frac{1}{2}D^*\gamma x^{*2} + O(x^{*3}) \tag{34}$$

$$exp[\mu(x^* - 1)] = e^{-\mu} + e^{-\mu}\mu x^* + \frac{1}{2}e^{-\mu}\mu^2 x^{*2} + O(x^{*3}) \tag{35}$$

Interestingly, if we slightly modified Eq. (34) by removing "1" from the brackets, then the Taylor expansion of Eq. (34) can be rewritten as:

$$\frac{D^*}{\gamma}[exp(x^*\gamma)] = \frac{D^*}{\gamma} + D^*x + \frac{1}{2}D^*\gamma x^{*2} + O(x^{*3}) \tag{36}$$

In this case, Eq. (35) and Eq. (36) are identical if they satisfy the following conditions:

$$D^* = e^{-\mu}\mu \tag{37}$$

$$\gamma = \mu \tag{38}$$

Thus, for given prior conditions (37) and (38), the main difference between the newly proposed model and Savenije's model lies in the inclusion of the term $-KD^*/\gamma = m\exp(-\mu)$ in the braces of Eq. (32), which is closely related to the upstream river discharge, dispersion coefficient at the estuary mouth, tidal frequency and the geometry of the estuary according to Eq. (16).

As an illustration, Figure 8 displays the longitudinal variations of the dimensionless salinity $S^*$ and its gradient $dS^*dx^*$ for different values of the recessing coefficient $\mu$ and the rising coefficient $m$, where the solid lines represents the solutions obtained by Savenije's model while the dashed lines obtained by the newly proposed model. It can be seen from Figure 8 that the additional term $m\exp(-\mu)$ mainly affects the downstream part of the salt intrusion curve, while the two methods tend to be the same for larger values of $x^*$ (at the toe of the salt intrusion curve).





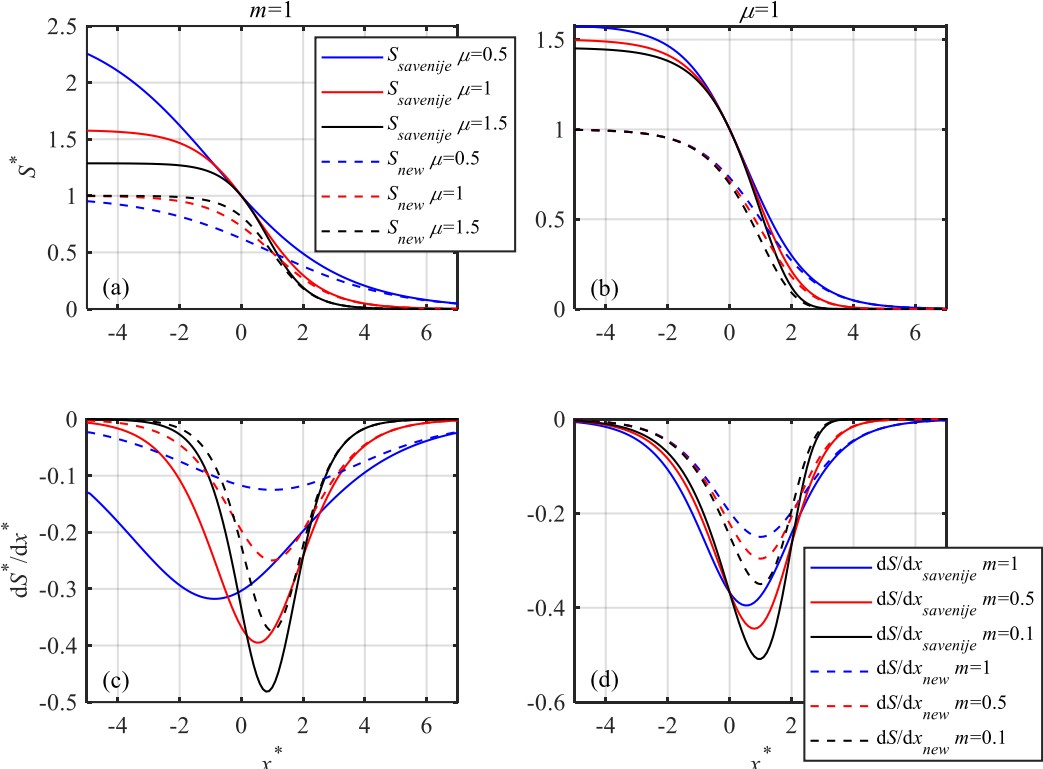

**Figure 8. Longitudinal variations of the dimensionless salinity $S^*$ (a, b) and its gradient $dS^*dx^*$ (c, d) along the estuary axis for different values of input parameters. The solid and dashed lines indicate the solutions obtained by Savenije's model and newly proposed model, respectively.**

It is worth noting that the above analysis suggests that the Van der Burgh's coefficient $K$ (being equal to $-m$) should be negative rather than positive since $m$ is generally positive (see Table 1). This indicates that the dispersion coefficient $D$ should be increased along the estuary axis according to Eq. (9). Figure 9 shows the analytically computed longitudinal dimensionless salinity $S^*$ using Eq. (18) for a wide range of the input parameters $K$ and $D^*$ when $\gamma=1$. We can see the computed $S^*$ does converge to 0 when the distance $x^*$ approaches infinity when $K$ values are negative, which is very different from the performance of previous analytical solutions using positive $K$ values, where the computed $S^*$ generally diverges for larger values of $x^*$. Consequently, from a curve-fitting perspective, the Savenije's model using negative $K$ values can be regarded as a special case of the newly proposed salt intrusion model if we further rescale the salinity by the dimensionless salinity in the deep ocean (see Figure S9 in the Supplementary Material). This also suggests that an enhanced empirical relationship concerning the effective dispersion coefficient (instead of the conventional Van der Burgh's relationship) is required for deriving an accurate salt intrusion curve from a theoretic point of view.

It should be noted that although the model fits the observations very well, the physical foundation of Eq. (28) needs further study in the future. This limitation could be relaxed by carefully comparing the proposed model with those based on the steady-




state salt balance equation. If the physics behind the Eq. (28) is understood, the model could be made fully predictive through the relation of the three model parameters ($x_p$, $\mu$ and $m$) to measurable or quantifiable variables (e.g., river discharge, tidal amplitude, cross-sectional area convergence length) by means of regression models or similar approaches. Moreover, the proposed salt intrusion model is particularly useful for quantifying the alterations in salt intrusion dynamics by comparing the three calibrated model parameters for two different periods with considerable different conditions owing to the climate change or human interventions.

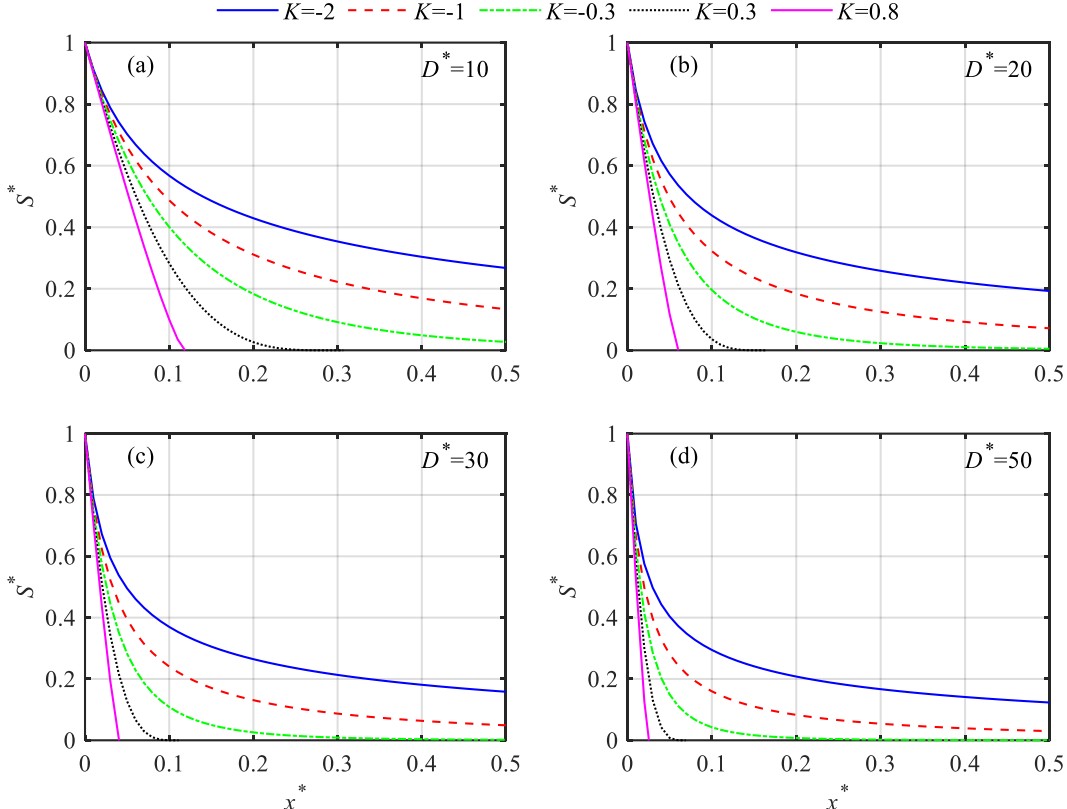

**Figure 9. Longitudinal variation of the dimensionless salinity $S^*$ computed using Eq. (18) along the estuary axis for different values of $K$ and $D^*$ when $\gamma=1$.**

## 5 Conclusions

In this paper, we revisited the empirical salt intrusion model making use of Guo's general unit hydrograph theory (2022a, 2022b, 2022c) and proposed a general and analytical model for the salinity distribution in estuaries of the partial to well mixed type. The newly developed method does not require observed or calibrated salinity at the estuary mouth and can be well calibrated using a minimum of three salt measurements along the estuary axis. In addition, the salinity converges towards zero as distance approaches infinity asymptotically, which might improve the model performance near the toe of the salt intrusion

curve when compared to empirical solutions based on salt balance equation. The model has been applied to numerous estuaries worldwide and the results agree very well with the observations. This indicates that the proposed model can be a useful tool to understand the dynamics of salt intrusion in estuaries and for assessing the potential impacts due to human-induced (e.g.,
dredging) or natural (e.g., mean sea level rise) changes. However, the underlying physical foundation of the proposed model and the physics of model parameters need further study in the future.

**Data availability**

All salt intrusion observations are available on the website at https://salinityandtides.com/data-sources/.

**Author contribution**

All authors contributed to the design and development of this work. The model was originally developed by Huayang Cai. Bo Li and Junhao Gu carried out the data analysis. Erwan Garel revised the paper. Tongtiegang Zhao reviewed the paper.

**Competing Interest**

The contact author has declared that neither they nor their co-authors have any competing interests.

**Acknowledgements**

This research has been supported by the National Natural Science Foundation of China (Grant No. 51979296), from the Guangdong Provincial Department of Science and Technology (Grant No. 2019ZT08G090), from the Guangzhou Science and Technology Program of China (Grant No. 202002030452). The first author very much appreciates the valuable suggestions from Prof. Guo Junke of University of Nebraska Lincoln in the derivations of Eqs. (20)-(30). The very valuable comments from Prof. Hubert Savenije of TU-Delft are very much appreciated. EG acknowledges the support of the Portuguese
Foundation for Science and Technology (FCT) through the grant UID/MAR/00350/2020 attributed to CIMA, University of Algarve.

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
