# Peer review of "Extension of the general unit hydrograph theory for the spread of salinity in estuaries"

_EGUsphere, 2023_

## Community Comment (CC1)

**Response letter to Reviewer#2**

We thank Dr. Daniel Thewes for the careful consideration of our work. We agree with his constructive and thoughtful comments and suggestions, which led to a much improved and complete manuscript. In this response letter, we have replied (in blue) to all the comments formulated by the Reviewer (in black).

**Major Comments:**

1. The topic and aim of the manuscript is clear from both the abstract and the introduction. It is of good relevance and can be a valuable contribution to its field. The overall quality of the text is good and easy to follow. The figures are of good quality, although some improvements can be made (see minor comments below).

Our reply: We very much appreciate all the comments and suggestions raised by the reviewer. In the revised manuscript, we shall completely address all the comments.

2. The manuscript would benefit from taking more research outside of the theoretical frame into account, such as modelling studies of estuaries, of which there are plenty. For instance, unstructured grid models can serve very well to demonstrate the authors theory in 3D model cases. For comparisons, see e.g. Pein et al., 2018, 10.1002/2016JC012623 (while the paper focusses on meandering and secondary effects, which are not directly relevant to the author's work, it gives an example of an idealised modelling study), and Eslami et al., 2019, 10.1038/s41598-019-55018-9. In chapter 4, the authors address the physical basis of equation 28, to which they rightfully say that future work is needed. It would be a good addition to the paper, if they expand on what sort of future work is necessary, and what past work is relevant. Generally, doing a more thorough comparison to related research can not only help to make the theoretical considerations more understandable, but can also increase the reach and impact of the paper.

Our reply: We agree with the Reviewer's comments. In the Discussion part, we shall

enhance the discussion concerning the related works and the necessary future works: "It should be noted that although the model fits the observations very well, the physical foundation of Eq. (29) needs further study in the future. This limitation could be relaxed by carefully comparing the proposed model with those based on the steady-state salt balance equation. **Specifically, several idealized numerical models (1-D, 2-D or 3-D models) have been adopted as a first approximation to quantify the along-channel salinity dynamics (e.g., Pein et al., 2018; Dijkstra and Schuttelaars, 2021; Wei et al., 2022). With these idealized numerical models, the physics behind the Eq. (29) can be understood that the model could be made fully predictive through the relation of the three model parameters ($x_P$, $\mu$ and $m$) to measurable or quantifiable variables (e.g., river discharge, tidal amplitude, cross-sectional area convergence length) by means of regression models or similar approaches. It should be noted that the proposed salt intrusion model in principle valid only for well mixed or partially mixed estuaries, where salt intrusion really matters. From a practical perspective, this is not a restrictive assumption since the salt wedge in highly stratified condition only occurs at the time of high river discharge, when flood protection is generally the main concern and salt intrusion is not relevant (Savenije, 2005, 2012).** Moreover, the proposed salt intrusion model is particularly useful for quantifying the alterations in salt intrusion dynamics by comparing the three calibrated model parameters for two different periods with considerably different conditions owing to the climate change or human interventions."

3. Not a very big point, but still potentially interesting: the authors correctly note in lines 196ff. that the salinity gradient is symmetric around x*=1 or x=x_p. They do not mention that S* is also centrally symmetric around the point (x*,S*)=(1,0.5). This is true, if m=1.

Our reply: We thank the reviewer for pointing this out. Indeed, substitution of $x^*=1$ into the Eq. (29) yields $S^*=0.5$ if $m=1$. In the revised manuscript, we shall explicitly mention that: "In addition, Figures 4c, d show that $m=1$ gives a symmetric salinity gradient about $x=x_P$ ($x^*=1$, $S^*=0.5$)".

4. Seeing as the authors and the reviewer agree that more work on the physical

foundation of the theory is needed, it may make sense to contextualise later, when μ and m are explained in further detail, what an estuary of such type may look like. In figure 7, and also figures S1-S8, they show different salt intrusion curves for estuaries of different shapes. The corresponding values of μ and m for those are found in table 1. Here it may make sense to speculate on how the shape of the estuary impacts these values.

Our reply: We very much appreciate the reviewer's suggestions. In the revised manuscript, we shall speculate on the relationship between the geometry of the estuary and the model parameters: "For illustrations, Figure 7 shows the longitudinal computation applied to the Pungue, Incomati and Limpopo estuaries by means of both the newly proposed and Savenije's models. Generally, the results of the two models are satisfactory for the different shapes of salt intrusion curves in well mixed or partially mixed estuaries (Savenije, 2005, 2012): (1) a "dome-shape" intrusion curve (such as Pungue, Figure 7a), which generally occurs in strong funnel-shaped estuaries; (2) a "bell-shaped" intrusion curve (such as Incomati, Figure 7b), which generally occurs in estuaries that have a trumpet shape ; (3) a "recession-shape" intrusion curve (such as Limpopo, Figure 7c), which generally occurs in narrow estuaries with a near-prismatic shape and a high river discharge. **It is worth noting that these types of salt intrusion curves are very much linked to the geometry of the estuary (Savenije, 2005, 2012). However, we observe from Table 1 that the calibrated model parameters ($\mu$ and $m$) are rather sensitive to the varied riverine and tidal forcing for a specific estuary. Thus, further studies concerning the relationship between the forcing conditions and the model parameters ($\mu$ and $m$) are required in the future.** It can be seen from Figure 7 that one important difference of performance between these two models lies in the rising limb when the distance approaches infinity. As $x$ tends to infinity, the salinity gradient of the newly proposed model asymptotically approaches to zero, while it reduces to zero at a critical position corresponding to the salt intrusion length from Eq. (20) for Savenije's model. This feature allows an improved fit with observations at the toe of the salt intrusion curve (e.g., Figure 7a). Figures S1-S8 show the comparison between the observed longitudinal salinity and the analytically computed

salt intrusion curves along 21 estuaries worldwide (see the Supplementary Material).".

5. One point that needs clarification is whether the method was tested on only those estuaries listed in table 1, or otherwise if there may have been estuaries in which the method failed to produce accurate results. There should be a paragraph on the limitations of application.

Our reply: Indeed, the limitation of the proposed model was detailed in the Discussion part: "It should be noted that although the model fits the observations very well, the physical foundation of Eq. (29) needs further study in the future. This limitation could be relaxed by carefully comparing the proposed model with those based on the steady-state salt balance equation. **Specifically, several idealized numerical models (1-D, 2-D or 3-D models) have been adopted as a first approximation to quantify the along-channel salinity dynamics (e.g., Pein et al., 2018; Dijkstra and Schuttelaars, 2021; Wei et al., 2022). With these idealized numerical models, the physics behind the Eq. (29) can be understood that the model could be made fully predictive through the relation of the three model parameters ($x_p$, $\mu$ and $m$) to measurable or quantifiable variables (e.g., river discharge, tidal amplitude, cross-sectional area convergence length) by means of regression models or similar approaches. It should be noted that the proposed salt intrusion model in principle valid only for well mixed or partially mixed estuaries, where salt intrusion really matters. From a practical perspective, this is not a restrictive assumption since the salt wedge in highly stratified condition only occurs at the time of high river discharge, when flood protection is generally the main concern and salt intrusion is not relevant (Savenije, 2005, 2012).** Moreover, the proposed salt intrusion model is particularly

useful for quantifying the alterations in salt intrusion dynamics by comparing the three calibrated model parameters for two different periods with considerably different conditions owing to the climate change or human interventions."

6. In Line 299ff., the authors claim that the model can be well calibrated using a minimum of three salt measurements along the estuary axis. While this is plausible (and visible in the supplement), it lacks explanation. Please expand on the mathematical basis for this claim. For instance, if the underlying equation was linear, one would need exactly two measurements, to give a slope and an intercept. Were it quadratic, three measurements would suffice, etc.. Equations 28 and 29 are not that simple though. While a curve fitting tool can make least square fits and give a result with just three measurements, it warrants some more theoretical insight.

For a start, equation 28 is monotonous, and equation 29 has precisely one minimum. Knowing that, it can be argued that three measurements of S can suffice, as one would know if one is to the left or the right of the gradient minimum.

There is no need for a larger discussion, yet, some explanations to the claim would be in order.

Our reply: We very much appreciate the reviewer's suggestions. In the revised manuscript, we shall explicitly mention that: "The newly developed method does not require observed or calibrated salinity at the estuary mouth and can be well calibrated using a minimum of three salt measurements along the estuary axis. **This is mainly due to the fact that Eq. (29) is a monotonic function that its first derivative (i.e., Eq. 30) does not change sign and has only one minimum.**"

**Minor and technical comments:**

1. Line 16: "longitudinal" can be misunderstood as meridional. Maybe chose a different word? "along-river", maybe? This is just a suggestion.

Our reply: We thank the review for pointing this out. In the revised manuscript, we shall

modify the sentence as: "we then derive a general unit hydrograph for the salinity distribution along the estuary of the partially to well mixed type."

2. Line 37 "[…] he derived a general and analytical unit-volume hydrograph for S-hydrograph, […]" – this is grammatically unclear.

Our reply: In the revised manuscript, we shall modify the sentence as: "he derived a general and analytical expression of the S-hydrograph in terms of a unit-volume of excess rainfall".

3. The paragraph, starting in line 42, appears to be a repetition of the abstract in some sense.

Our reply: Indeed, here we would like to clarify the organization of the manuscript, which is kind of repetition of the abstract.

4. The unit nomenclature is slightly confusing. Perhaps reformat [T-1] to [1/T], or write the "-1" in superscript. In line 97ff., it is in superscript, so I presume this is an error.

Our reply: You are right. It should be $T^{-1}$. In the revised manuscript, we shall correct this mistake: "It was shown that a classical instantaneous UH $u(t)$ [$T^{-1}$] (representing the discharge due to a unit-volume input of excess rainfall) with regard to time $t$ [T] should satisfy the following properties (Chow et al., 1988)".

5. Equation numbers are wrong.

- There are two equations (1). In the text, equation (2) and (4) are referenced (line 59), but appear to correspond to equations (1) – the second one – and (3), respectively. This error continues, going forward.

- the caption to figure 1 references equation (5), while the text references (6) as the same equation.

- In later chapters, the numbers sometimes match the text and sometimes they do not. Please pay extra attention here.

Our reply: Many thanks for pointing this out. In the revised manuscript, we shall correct this mistake.

6. Line 74: an article missing in front of "S-hydrograph", same in line 78.

Our reply: Thanks a lot for pointing this out. In the revised manuscript, we shall supplement the missing "the" in front of "S-hydrograph".

7. "t_p" is only introduced in line 81, but might also be reflected in figure 1 and 2, for extra clarity.

Our reply: We very much appreciate the reviewer's suggestion. In the revised manuscript, we shall update the Figure 2 as follows.

[Figure]

Fig. 2. Illustration of the S-hydrograph and instantaneous UH analytically computed using Eqs. (7) and (9), respectively, for given values of $\beta_1=\beta_2=3$ and $t_p=10$.

8. Line 92: "deriving", not "derive"

Our reply: We shall correct this mistake as suggested.

9. Line 97: "landward" is confusing here. Maybe say "upstream"?

Our reply: In the revised manuscript, we shall replace "landward" with "upstream".

10. Line 143: kg/m³ is an uncommon unit for salinity. g/kg is more common.

Our reply: In this study, we would prefer to use the International System of Units (SI), which is consistent with those used in Savenije (2005, 2012).

11. Line 148-151: plausible, but could be moved to the discussion.

Our reply: We agree with this comment. In the revised manuscript, we shall move these sentences to the discussion part.

12. Line 171 is misleading: μxp->μ would imply that xp=1. Consider changing one of the μ to something like μ_n or μ*, or whichever.

Our reply: We agree with this comment. In the revised manuscript, we shall use $\mu x_\mathrm{p} \rightarrow \mu^*$.

13. Line 191: technically, we see a decrease, not an increase in salinity gradient. The absolute or the gradient magnitude increases. Best rephrase that sentence.

Our reply: You are right. In the revised manuscript, we shall correct this sentence: "For $x^*<1$ we see an exponential decrease of salinity gradient until a minimum value is reached at a critical position $x=x_\mathrm{p}$ (or $x^*=1$) defined by Eq. (27), beyond which the salinity gradient decreases exponentially until zero is reached asymptotically (Fig. 4c, d)."

14. Line 193: "… is reached asymptotically (fig. 4c, d)."

Our reply: Many thanks for the suggestion.

15. Figure 5 has y-limits of 0.5 and 5, yet table 1 has values of μ from 0 to 6. m goes from 0.1 to ~6. L is regularly larger than 18 in the table. Consider adapting figure 5 in such a way that it covers the table's values.

Our reply: Many thanks for the suggestion. Note that the $L^*$ represents the dimensionless salt intrusion length that is scaled by the inflection point $x_p$. In the revised manuscript, we shall update the Figure 5 as follows:

[Figure]

Fig. 5. Response of dimensionless salt intrusion length $L^*$ to the dimensionless parameters $\mu$ and $m$ for a given salinity threshold $S_f^*=0.01$.

16. Lines 229ff: While figure 7a does show the Pungue estuary, Incomati is 7c and Limpopo 7e.

Our reply: In the revised manuscript, we shall correct these mistakes.

17. Figure 7: add x-label. Typically, distance along rivers is measured from source to sea. It therefore makes sense to state, explicitly, that this is sea to source.

Our reply: Many thanks for the suggestion. In the revised manuscript, we shall update the Figure 7 as follows:

[Figure]

Fig. 7. Observed and analytically computed salt intrusion curves using the newly proposed and Savenije's models in the Pungue estuary (a, b), in the Incomati estuary (d, e) and in the Limpopo estuary (g, h), together with the idealized shape of the estuary (c: Funnel shape; f: Trumpet shape; i: Prismatic shape).

18. Again, lines 229ff.: the shapes of the estuaries are named in the text (only explained in Savenije, 2012), but they could be shown in idealised form in a third column of figure 7, i.e., draw a funnel, a trumpet and a prism shape. This can be very rough and idealised.

Our reply: Many thanks for the suggestion. In the revised manuscript, we shall update the Figure 7 (see figure above).

19. Line 252: "it is difficult to compare eq. 31 and eq 33 directly", or "… to directly compare …"

Our reply: In the revised manuscript, we shall revise the sentence as suggested.

20. Equation 26: missing asterisk (*) for the x* in the linear term (second on the RHS).

21. Line 265: dS*/dx* (missing "/")

22. Figure 8 and line 267: why are dashed and solid lines switched, with respect to figure 7? Consistency between the figures would be favourable.

[Figure]

Fig. 8. Longitudinal variations of the dimensionless salinity $S^*$ (a, b) and its gradient $\mathrm{d}S^*\mathrm{d}x^*$ (c, d) along the estuary axis for different values of input parameters. The solid and dashed lines indicate the solutions obtained by Savenije's model and newly proposed model, respectively.

23. Line 291: considerably, not considerable, same in 298: partially, not partial

Our reply: Thanks a lot for pointing this out. In the revised manuscript, we shall correct these mistakes.

References:

Dijkstra, Y. M., & Schuttelaars, H. M. (2021). A unifying approach to subtidal salt intrusion modeling in tidal estuaries. Journal of Physical Oceanography, 51(1), 147–167. https://doi.org/10.1175/jpo-d-20-0006.1.

Pein, J., Valle-Levinson, A., & Stanev, E. V. (2018). Secondary circulation asymmetry in a meandering, partially stratified estuary. Journal of Geophysical Research: Oceans, 123, 1670–1683. https://doi.org/10.1002/2016JC012623.

Savenije, H.H.G., 2005. Salinity and tides in alluvial estuaries. New York: Elsevier. https://doi.org/10.1016/B978-0-444-52107-1.X5000-X.

Savenije, H.H.G., 2012. Salinity and tides in alluvial estuaries. completely revised 2nd edition. Available from http://www.salinityandtides.com [Accessed 22 March 2022].

Wei, X., Williams, M. E., Brown, J. M., Thorne, P. D., & Amoudry, L. O. (2022). Salt intrusion as a function of estuary length in periodically weakly stratified estuaries. Geophysical Research Letters, 49, e2022GL099082. https://doi.org/10.1029/2022GL099082.